



# Challenges in retrieving stratospheric aerosol extinction and particle size from ground-based RMR-LIDAR observations

Jacob Zalach[1], Christian von Savigny[1], Arvid Langenbach[2], Gerd Baumgarten[2], Franz-Josef Lübken[2], and Adam Bourassa[3]

[1]Institute of Physics, Greifswald University, Felix-Hausdorff-Str. 6, 17489 Greifswald, Germany
[2]Leibniz-Institute of Atmospheric Physics, Schlossstr. 6, 18225 Kühlungsborn, Germany
[3]University of Saskatchewan, 116 Science Place, Saskatoon, SK S7N 5E2, Canada

*Correspondence to:* J. Zalach (zalachj@uni-greifswald.de)

**Abstract.** We report on the retrieval of stratospheric aerosol particle size and extinction coefficient
profiles from multi-color backscatter measurements with the Rayleigh-Mie-Raman lidar operated at
the Arctic Lidar Observatory for Middle Atmosphere Research (ALOMAR) in northern Norway.
The retrievals are based on a two-step approach. In a first step the median radius of an assumed log-
normal particle size distribution with fixed width is retrieved based on the color ratio formed from
the measured backscatter ratios at wavelenghts of 1064 nm and 532 nm. An intrinsic ambiguity of
the retrieved aerosol size information is discussed. In a second step, this particle size information is
used to convert the measured lidar backscatter ratio to aerosol extinction coefficients. The retrieval
is currently based on monthly-averaged lidar measurements covering the period from the year 2000
to present. A sensitivity study is presented that allows establishing an error budged for the aerosol
retrievals. Assuming a log-normal aerosol particle size distribution with a geometric width of S=1.3,
median radii on the order of 100 nm are retrieved. The median radii are found to generally decrease
with increasing altitude. The retrieved aerosol extinction profiles are compared to observations with
various current and past satellite instruments.

## 15 1   Introduction

### 16 1.1   Importance of stratospheric aerosols

Stratospheric aerosols are of crucial importance for various physical and chemical processes in the
Earth's atmosphere. According to the current understanding, the main component of stratospheric
aerosols consists of sub-micron particles made of $H_2SO_4$ and $H_2O$. This stratospheric sulfate aerosol





layer is thought to be maintained under volcanically quiescent conditions by a continuous influx of
OCS from the troposphere (Crutzen, 1976). Volcanic eruptions may lead to stratospheric injections
of sulfur compounds, which can be photochemically converted to $H_2SO_4$ and contribute to the for-
mation and growth of sulfate aerosol particles. Stratospheric sulfate aerosols scatter incoming solar
radiation and also absorb and re-emit terrestrial thermal radiation. The net effect of an enhanced
stratospheric sulfate aerosol loading is generally a surface cooling.
Stratospheric sulfate aerosols also provide surfaces for heterogeneous chemical reactions. For an
anthropogenically enhanced stratospheric halogen loading, an increase of the aerosol surface area
leads to a net catalytic destruction of stratospheric $O_3$. This effect will be reversed, once the strato-
spheric halogen load has returned to background levels (e.g. Tie and Brasseur, 1995). In polar re-
gions stratospheric aerosols provide condensation nuclei for polar stratospheric clouds (PSC) which
facilitate heterogeneous chemical reactions that lead to chlorine activation, which in turn leads to
catalytic ozone loss.

### 33   1.2   Novel aspects of the utilised approach

Up to now long-term lidar observations of stratospheric aerosols were essentially limited to mid-
latitudes. The atmosphere in the Arctic region is much less explored and exhibits characteristic
variations of its aerosol load as well as its thermal, dynamic and chemical properties. In the cur-
rent study observations of stratospheric aerosols obtained by measurements with the Rayleigh-Mie-
Raman-lidar (RMR-lidar) at the ALOMAR (Arctic Lidar Observatory for Atmospheric Research)
station (von Zahn et al., 2000) located at 16.0° E, 69.3° N are analysed.
The number of experimental studies on the size of stratospheric sulfate aerosols is quite limited
and the published aerosol sizes cover quite a large range of values, even under background aerosol
conditions. A major advantage of this new method is that the lidar ratio does not have to be assumed,
but is calculated from the measurements themselves. For most other lidar studies on stratospheric
aerosols the value of the lidar ratio is determined based on a priori assumptions of the aerosol particle
size distribution. In addition, the lidar ratio is usually assumed to be independent of altitude, which
is generally not true.
To our best knowledge, the approach employed here – i.e. the retrieval of aerosol particle size
information in a first step, followed by calculating extinction coefficients – has not yet been applied
to lidar measurements of stratospheric sulfate aerosols. A similar approach, however, has been
employed by Blum et al. (2006) and Jumelet et al. (2008) for investigating polar stratospheric clouds
(PSCs).

### 52   1.3   Outline

The paper is structured as follows. Section 2 provides a brief description of the lidar system whose
measurements are used in the present study. In section 3 we describe the steps of the retrieval ap-





proach employed to, first, obtain information on the aerosol particle size and, second, to calculate
the lidar ratio and retrieve the aerosol extinction coefficient and number density profile. The re-
trieval errors are discussed in section 4 and the inferred aerosol extinction coefficient profiles are
compared with satellite-borne occultation and limb-scatter measurements in section 5. Conclusions
are presented in the final section 6.

## 2   The ALOMAR Rayleigh-Mie-Raman (RMR) lidar


The RMR lidar is described in von Zahn et al. (2000) and the basic data processing steps are found
in Brand et al. (2019). It has the ability to measure elastically scattered photons at its principal
wavelengths of 1064 nm, 532 nm and 355 nm, which are scattered by molecules and aerosol particles,
as well as Raman-scattered photons at 387 nm and 608 nm, which are scattered by molecules only.
This feature allows to calculate backscatter ratios for the principal wavelengths given above.
The backscatter ratio for a given wavelength is the ratio of the detected signal originating from
scattering processes on aerosol particles (Mie-scattering) and air molecules (Rayleigh-scattering)
normalized by the contribution of molecular scattering only (see eq. (3)). Therefore it contains
information about the aerosol load in the scattering air volume. These backscatter ratios will be used
here for the retrieval and further explained in the next section.

## 3   Retrieval approach


The main goal of this study is to describe an approach for the retrieval of vertical extinction and
particle size profiles of stratospheric sulfate aerosol from ground-based multi-color lidar observa-
tions. Since backscatter ratios at three wavelengths are available, a method to simultaneously infer
the log-normal distribution width and median radius from two color ratios could in principle be em-
ployed as described by von Cossart et al. (1999) and Baumgarten et al. (2007), who applied it to lidar
measurements of noctilucent clouds (NLCs). This method is based on the exploitation of two color
ratios determined from the lidar measurements at the three available wavelengths (355 nm, 532 nm
and 1064 nm). The method works well for the relatively small NLC particles which reach radii of
up to about 100 nm, but is not generally applicable to stratospheric aerosol particles, whose median
radii may well exceed 100 nm. Therefore, this approach is not utilized here. Instead, a simplified
approach, as outlined by Yue and Deepak (1983), is necessary which sets one of the two distribution
parameters to a constant value (here, the distribution width) and retrieves the remaining one (the
median radius).
The retrieval is performed in two steps. First, aerosol particle size is found by comparing the
measured and modelled color ratio of the backscatter ratio profiles at two different wavelengths.
This can be done because the color ratio for the wavelengths used depends on the median radius of
the assumed log-normal size distribution. Second, the inferred particle size is employed to calcu-



late aerosol extinction coefficient profiles from the measured backscatter ratio profiles. Finally, the
aerosol particle density is determined, once particle size and extinction coefficient are known.

### 3.1   Retrieval assumptions

Some general assumptions were made prior to the retrieval.  Backscatter ratios at 1064 nm and
532 nm were chosen for the retrieval instead of 355 nm, because measurements at this wavelength
essentially serve as a measure for Rayleigh scattering (Brand et al., 2019). The aerosol is assumed
to consist of 75% sulfuric acid ($H_2SO_4$) and 25% water which defines its refractive index. For the
assumed aerosol composition the real part of the refractive index is roughly 1.43 at the wavelength
of 532 nm and 1.42 at 1064 nm (Palmer and Williams, 1975). Aerosol absorption turned out to have
only negligible influence on the retrieval, therefore the imaginary part of the refractive index was set
to zero. The particle size distribution (PSD) is assumed to be log-normal,

$$\frac{dN_A}{dr} = \frac{N_A}{\sqrt{2\pi}\ln(S)r_m} \cdot exp\left(-\frac{(\ln(r)-\ln(r_m))^2}{2\ln^2(S)}\right) \tag{1}$$

with $N_A$ as the number density of the aerosol particles, $S$ the geometric standard deviation (distri-
bution width) and $r_m$ the median radius. Here, the assumption of the distribution width becomes
important. Since its value often lies in the range between 1.2 and 1.4 (e.g. Bingen et al., 2004a), a
value of $S = 1.3$ was chosen. We note that this value is somewhat arbitrary and has a direct impact
on the values of the retrieved median radii. However, the impact on the inferred aerosol extinction
coefficients is relatively weak (see section 4). Distribution widths other than 1.3 will only be used to
determine the impact of a possible inaccurate assumption of this parameter on the retrieval.

### 3.2   Retrieval of aerosol particle size information

In the first step the particle size is retrieved, which is a necessary requirement for further computa-
tions. The lidar backscatter ratio $R(z,r_m,\lambda)$ at altitude $z$ and wavelength $\lambda$ is given by

$$R(z,r_m,\lambda) = \frac{\beta_{Mie}(z,r_m,\lambda) + \beta_{Ray}(z,\lambda)}{\beta_{Ray}(z,\lambda)} \tag{2}$$

which can be simplified to

$$R(z,r_m,\lambda) = \frac{\beta_{Mie}(z,r_m,\lambda)}{\beta_{Ray}(z,\lambda)} + 1 \tag{3}$$

with the aerosol and Rayleigh volume backscatter coefficients $\beta_{Mie}(z,r_m,\lambda)$ and $\beta_{Ray}(z,\lambda)$ which
are defined as

$$\beta_{Mie}(z,r_m,\lambda) = k_{Mie}^{sca}(z,r_m,\lambda) \cdot P_{Mie}(\Theta,r_m,\lambda) \tag{4}$$

and





$$\beta_{Ray}(z,\lambda) = k^{sca}_{Mie}(z,\lambda) \cdot P_{Ray}(\Theta) \tag{5}$$

Here, $k^{sca}_{Mie/Ray}$ denotes scattering coefficients, $P_{Mie/Ray}$ the phase functions for Mie and Rayleigh
scattering, respectively and $\Theta = 180°$ is the scattering angle. The scattering coefficients $k^{sca}_{Mie/Ray}$
depend on air/aerosol densities and the respective scattering cross sections. In the case of Mie
scattering on aerosol particles eq. (4) can be expanded to include both values giving

$$\beta_{Mie}(z,r_m,\lambda) = N_A(z) \cdot \sigma_{Mie}(z,r_m,\lambda) \cdot P_{Mie}(\Theta,r_m,\lambda) \tag{6}$$

with $N_A(z)$ as the aerosol particle density. For Rayleigh scattering an analogous relationship is
true

$$\beta_{Ray}(z,\lambda) = N(z) \cdot \sigma_{Ray}(z,\lambda) \cdot P_{Ray}(\Theta). \tag{7}$$

All values used are summarized and explained in Table 1.

| | |
|---|---|
| $\Theta$ | Scattering angle ($\Theta = \pi$ for lidar observations) |
| $R$ | Backscatter ratio (dimensionless) |
| $k^{sca}_{Mie}$ | Mie scattering coefficient [km$^{-1}$] |
| $k^{sca}_{Ray}$ | Rayleigh scattering coefficient [km$^{-1}$] |
| $P_{Mie}(\Theta,\lambda)$ | Mie scattering phase function [sr$^{-1}$] |
| $P_{Ray}(\Theta)$ | Rayleigh scattering phase function [sr$^{-1}$] |
| $\beta_{Mie}$ | Mie volume scattering coefficient [km$^{-1}$ sr$^{-1}$] |
| $\beta_{Ray}$ | Rayleigh volume scattering coefficient [km$^{-1}$ sr$^{-1}$] |
| $N_A(z)$ | Aerosol particle density [m$^{-3}$] |
| $N(z)$ | Air (molecule) density [m$^{-3}$] |
| $\sigma_{Mie}(z,\lambda)$ | Mie scattering cross section [m$^{-2}$] |
| $\sigma_{Ray}(z,\lambda)$ | Rayleigh scattering cross section [m$^{-2}$] |

**Table 1.** Compilation of used variables and terminology.


Using lidar backscatter ratio measurements at two different wavelengths, a color ratio $C$ can be
formed

$$
\begin{aligned}
C(z,r_m,\lambda_1,\lambda_2) &= \frac{R(z,r_m,\lambda_1) - 1}{R(z,r_m,\lambda_2) - 1} \\
&= \frac{\beta_{Mie}(z,r_m,\lambda_1)}{\beta_{Mie}(z,r_m,\lambda_2)} \cdot \frac{\beta_{Ray}(z,\lambda_2)}{\beta_{Ray}(z,\lambda_1)}
\end{aligned}
\tag{8}
$$

with $\lambda_1$ = 1064 nm and $\lambda_2$ = 532 nm. Considering the last factor of eq. (8) – describing the
contribution of Rayleigh scattering at the two wavelengths – all parameters cancel out which are



present in both the numerator and denominator. Considering eq. (7) this is obviously the case for the
Rayleigh scattering phase function $P_{Ray}(\Theta)$ as well as the vertical air density profile $N(z)$. Thus,
the contribution of Rayleigh scattering at the two wavelengths in eq. (8) is reduced to a ratio of
Rayleigh scattering cross sections (Bucholtz, 1995):

$$\sigma_{Ray}(\lambda) = \frac{24\pi^3}{\lambda^4 N_0^2} \frac{(n_0(\lambda)^2 - 1)^2}{(n_0(\lambda)^2 + 2)^2} \left( \frac{6 + 3\gamma(\lambda)}{6 - 7\gamma(\lambda)} \right). \tag{9}$$

Here, $N_0$ denotes the number density of air for standard conditions, $n_0(\lambda)$ is the refractive index for
the given wavelength, also for standard conditions, with $n_0(532nm) = 1.0002782$, and $\gamma(\lambda)$ is the
depolarisation factor of air. Finally, the color ratio from eq. (8) becomes

$$C(z, r_m, \lambda_1, \lambda_2) = \frac{\beta_{Mie}(z, r_m, \lambda_1)}{\beta_{Mie}(z, r_m, \lambda_2)} \cdot \left( \frac{\sigma_{Ray}(\lambda_1)}{\sigma_{Ray}(\lambda_2)} \right)^{-1}. \tag{10}$$

Beside the dependence on altitude and wavelength the color ratio in eq. (10) depends only on
the assumed PSD and the aerosol refractive index which implicitly affects the aerosol backscatter
coefficient.
For the forward model a log-normal PSD was assumed (eq. (1)). As already mentioned, our
retrieval approach uses a single color ratio from measurements at 1064 nm and 532 nm, so only a
single particle size parameter can be retrieved, e.g., the median radius of an assumed PSD with fixed
width. In this study, the standard deviation of the PSD is set to a fixed value of $S = 1.3$ and the
median radius $r_m$ is retrieved using eq. (10).
The Mie-scattering cross sections $\sigma_{Mie}(z, r_m, \lambda)$ and Mie phase functions $P_{Mie}(\Theta, \lambda, r_m)$ used in
the forward model (eq. (10) and (4)) are calculated using the `IDL` Mie scattering routines provided
by the University of Oxford (Mie scattering routines, 2018).
Finally, by evaluating eq. (10) the color ratio is obtained as a function of radius $r_m$ as shown in
Figure 1 for the wavelengths 1064 nm and 532 nm and for different distribution widths. Then, color
ratios obtained from measured data are directly compared with the computed curve. This comparison
allows to identify those radii for which the curve has the same value as the measurement.

### 147 3.3 Radius ambiguity

The proper radius assignment described above requires further explanations as it is not necessarily
unique. As seen in Figure 1 most of the measured color ratio values can be found at several points
on the color ratio-radius curve. This behaviour depends on the assumed distribution width. To give
an example, we assume a measured color ratio value of 3 as marked in the Figure (dashed horizontal
line) and a distribution width of $S = 1.1$ (black curve). The color ratio function gives the same value
for median radii of roughly 100 nm, 270 nm and 310 nm. With respect to its local extrema the curve
can be divided into intervals which can be labeled as branch 1, branch 2, branch 3 and so on. For
example, the first branch would extend from the first local minimal value at r=0 nm to r=146 nm,
whereas branch 3 would range from roughly r=290 nm to r=400 nm.



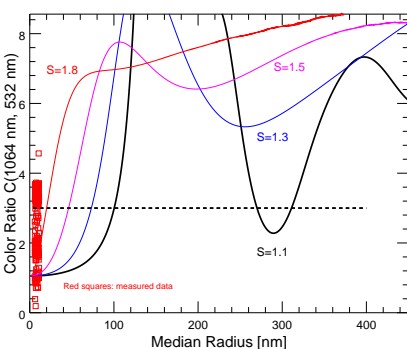

**Fig. 1.** Radius dependence of the color ratio $C$ for $\lambda_1 = 1064$ nm and $\lambda_2 = 532$ nm (see eq. (8)) calculated using a Mie scattering code for different distribution widths $S$ (solid lines). As an example, color ratios from measured monthly averaged backscatter profiles of March 2013 are shown as open red squares on the left.

The division into separate branches is important to illustrate the ambiguity of the radius retrieval
employing this method. Since the aerosol radius is not known in advance, the correct branch for the
retrieval has to be chosen using physical considerations.
First, it should be noted that by increasing the assumed distribution width the computed color
ratio curve changes its shape. This change has two consequences – branch 1 shifts toward smaller
radii and the eye-catching minimum at roughly 290 nm for $S = 1.1$ rises fast. This rise leaves an
ever growing portion of measured data points below its minimal value, hence without a possibility to
assign a radius using other branches than branch 1. Above a distribution width of around $S = 1.75$
any ambiguity vanishes since no minima are found on the computed color ratio curve. A subdivision
into several branches is then not possible anymore. Therefore only branch 1 allows for a radius
retrieval which covers the whole altitude range for all possible distribution widths. In this context it
is worth to point out that several studies report distribution widths well exceeding 1.4 (McLinden et
al., 1999; Bourassa et al., 2008; Ugolnikov et al., 2018). Some studies report $S = 1.1$ and even lower
(Bingen et al., 2004a), but we think that these values are not compatible with our measurements: for
aerosol populations with a rather small particle size only a retrieval based on branch 1 gives plausible
results since it allows for a smooth transition between very small radii and the maximal radius.
Retrievals based on other branches lead to a radius distribution with a very high minimal value of
several hundred nanometer without any transition to smaller values along the vertical profile. Figure
2 shows the issue. For color ratios of $C < 2.4$ (see Figure 1), branch 3 reproduces the measured color
ratio only for distribution widths smaller than $S = 1.1$. In this cases the particle size would be larger
than 290 nm. We see two reasons for excluding branch 3: Very small distribution withs with $S \leq 1.1$
are not compatible with our data set as we use monthly mean lidar data. The variability of the

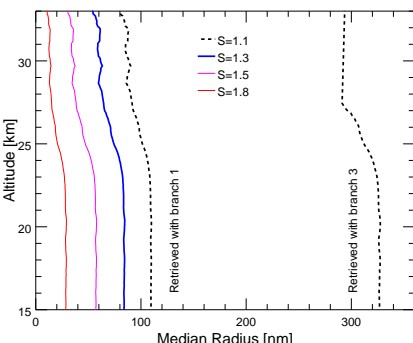

**Fig. 2.** Retrieved median radius profiles based on branch 1 (curves on the left) and branch 3 (black curve on the right). The branch 1 retrievals are shown for different $S$ to illustrate its influence on the radius retrieval. For $S < 1.1$ a radius assignment using branch 3 exists for the altitude range up to 30 km as shown in the rightmost profile (for $S = 1.1$). Shown are monthly mean profiles for March 2013.

background conditions throughout the month should lead to distribution widths $S \geq 1.1$. Secondly,
the absence of median radii below some 100 nm does not characterise a physically plausible aerosol
size profile.
The other radius profiles shown in Figure 2 were obtained by evaluating branch 1 of the color
ratio function for different distribution widths. In the shown example the radius retrieval evaluating
branch 2 only was discarded because the resulting radius profile is inverted with a steady growing
median radii with altitude. So branch 2 leads to obviously implausible profiles.
The other branches of the color ratio curve become important, if the particle size distribution
extends to median radii beyond the local extrema of the curve (Figure 1). This may happen, for
example, after volcanic eruptions (Deshler, 2008). In such cases the retrieval using single branches
of the color ratio function would recover only parts of the whole profile. This feature is a weak
point of the method employed here and reduces its robust application to aerosol populations with
radii below roughly 150 nm. However, such cases like volcanic eruptions are usually identified by
sudden enhancements of the backscatter ratio at limited altitudes and may be removed from further
processing (Brand et al., 2019).
**3.4  Derivation of extinction profiles**
Once the radius is determined the extinction profile is calculated. To compute the desired extinction
profiles eq. (3) can be solved for

$$k_{Mie}^{sca}(z, r_m, \lambda) = \frac{k_{Ray}^{sca}(z, \lambda) \cdot P_{Ray}(\Theta)}{P_{Mie}(\Theta, r_m, \lambda)} \cdot (R(z, r_m, \lambda) - 1). \tag{11}$$

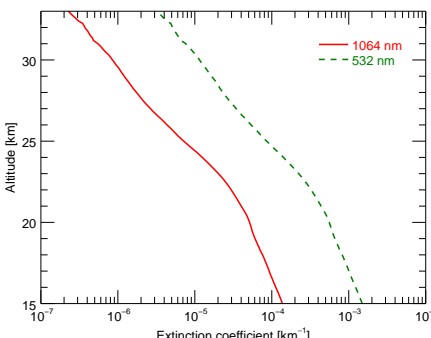

**Fig. 3.** Monthly mean of extinction coefficient profiles for March 2013 retrieved for an assumed distribution width of $S = 1.3$.

All quantities on the right hand side of this equation are either measured, like the backscatter ratio
$R(z,\lambda)$, or can be retrieved. The air density is calculated from monthly means (of daily means) of
temperature and pressure profiles obtained from ERA-Interim data sets provided by the European
Centre for Medium-range Weather Forecast (ECMWF Database, 2018). With those ingredients in-
serted into eq. (11) extinction profiles are computed for the two wavelengths initially used to derive
the particle size distribution. Figure 3 shows as an example the aerosol extinction profiles at 532 nm
and 1064 nm based on monthly averaged lidar measurements for March 2013 and assuming a distri-
bution width of $S = 1.3$.
The calculation in eq. (11) makes use of the reciprocal value of the Mie phase function for a
scattering angle of $180°$, the so called lidar ratio $\Lambda$, which is defined as the ratio of the extinction
and backscatter coefficient

$$\Lambda(r_m,\lambda) = \frac{k_{sca}(r_m,\lambda)}{\beta_{Mie}(\Theta,r_m,\lambda)} = \frac{1}{P_{Mie}(\Theta,r_m,\lambda)}. \tag{12}$$

The possibility to compute the Mie phase function, and therefore the lidar ratio, from the retrieved
median radius is an advantageous feature of this method.
In cases where a computation like in our approach is not possible usually a constant lidar ratio
with values around $50\,\mathrm{sr}$ for 532 nm is often assumed, e.g. Khaykin et al. (2017). To compute the
extinction coefficient this way, eq. (12) is used leading to

$$k_{Mie}^{sca}(z,\lambda) = \Lambda(r_m,\lambda) \cdot \beta_{Mie}(\Theta,z,\lambda). \tag{13}$$

This approach constitutes the only way to estimate aerosol extinction coefficients if lidar measure-
ments are available only at a single wavelength. However, assuming a constant lidar ratio does not

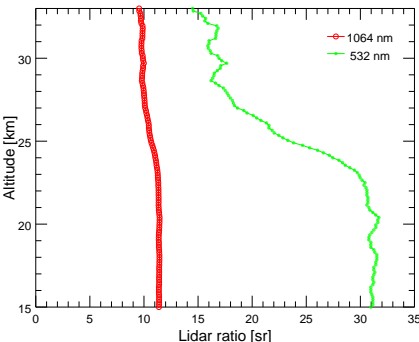

**Fig. 4.** Dependence of the lidar ratio on median radius for an assumed distribution width of S=1.3.

account for its dependence on the radius, and therefore on its implicit dependence on altitude. The
lidar ratio profile in Figure 4 shows that the assumption of a constant lidar ratio can be an good ap-
proximation for a certain altitude range, here between 15 and 23 km. But for altitudes above 23 km
the lidar ratio changes significantly, therefore leading to inevitable errors if its value is assumed con-
stant. To give an example, if our retrieval is done with a constant lidar ratio of 30 sr for 532 nm – this
value is taken from Figure 4 – the resulting extinction coefficient profile agrees very well with our
approach desctibed in this work for altitudes below 23 km but the deviation rises to roughly 80% at
30 km. This deviation would be bigger if a higher value (of the constant lidar ratio) would be chosen,
e.g. roughly 300% for 50 sr.
**3.5   Particle number density estimation**
Together with the extinction profiles retrieved in the previous section all information is available
to estimate the aerosol number density $N_A$ utilising the relationship between extinction coefficient,
scattering cross section and particle number density. With the mean cross section of a given PSD

$$\langle \sigma_{Mie}(z,\lambda) \rangle = \int\limits_0^\infty \frac{dN_A(r,r_m(z),S)}{dr} \cdot \sigma_{Mie}(r,\lambda,n_0^A) \, dr, \tag{14}$$

with $n_0^A$ as the refractive index of the aerosol, the particle density is given by

$$N_A(z) = \frac{k_{Mie}^{sca}(z,\lambda)}{\langle \sigma_{Mie}(z,\lambda) \rangle}. \tag{15}$$

For both extinction profiles retrieved one step earlier (eq. (11)) density profiles are computed
which, of course, are identical for the two wavelengths as can be seen in Figure 5.



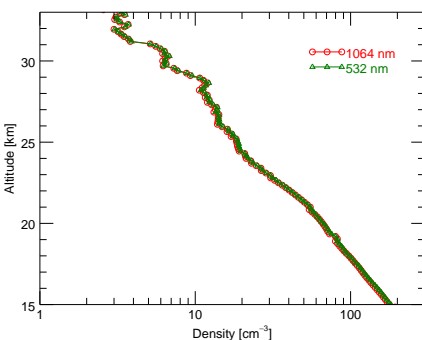

**Fig. 5.** Stratospheric aerosol particle density profiles obtained from the extinction coefficient profiles at 532 nm and 1064 nm shown in Figure 3. Monthly mean for March 2013 with an assumed distribution width of $S = 1.3$.

## 4  Error estimation

A key point of the retrieval approach employed here is a correct radius determination, since all inferred quantities depend on it. Therefore, the input parameters, i.e. the assumed distribution width ($S$) and the aerosol refractive index ($n_0$), are carefully chosen and their impact on the result – together with impact of measurement errors – is assessed in order to obtain a measure of the reliability of the results.

In order to determine the impact of every parameter on the resulting aerosol particle size and extinction coefficients, the retrieval was performed with consecutively perturbed parameter values used in the forward model. The following perturbations from nominal values were assumed: a temperature perturbation of $\pm 1\,K$, a pressure perturbation of $\pm 1\%$, a refractive index perturbation of $0.04$ (which corresponds roughly to a 20% change of the $H_2SO_4$ concentration) and finally a distribution width perturbation of $\Delta S = \pm 0.1$ and $\Delta S = \pm 0.2$, respectively.

For the first step of the retrieval procedure, i.e. the radius determination, the error contribution of each parameter is shown in Figure 6. Since the radius is derived from the computed color ratio it does not depend on temperature and pressure because those values cancel out (see eq. (10)). The most significant impact on the radius determination comes from an incorrect assumption on the distribution width. A difference of $\Delta S = \pm 0.2$ leads to relative error of slightly below $\Delta r/r = \mp 40\%$. If the assumed distribution width is increasing, then the retrieved median radius is decreasing, and vice versa. If the single error contributions are simply added to a total error – separately for the two assumed deviations of the distribution width – an absolute error range can be assigned to the retrieved radius profile as shown in Figure 7. The contribution of every parameter variation to the total error of aerosol extinction coefficients is shown in Figure 8. Again, the dominant contribution





comes from an erroneous assumption of the distribution width. This contribution is, however, not as
big as in the case of the radius retrieval shown in Figure 6 because the impact on the Mie phase func-
tion – which is needed to compute the extinction coefficient (eq. (11)) – associated with a high/low
bias in the distribution width $S$ is partly compensated by a low/high bias in the retrieved value of
$r_m$. Therefore, the inferred extinction coefficients seem to be rather robust against variations of the
distribution width as seen in Figure 9, which shows extinction coefficient profiles together with the
total accumulated error. Like in Figure 7 two error ranges are given for the two assumed errors of
the distribution width of $\Delta S = \pm 0.1$ and $\Delta S = \pm 0.2$, respectively.
Finally, we investigate the error of the number density in Figure 10. Due to its direct depen-
dency on the distribution width and radius (eq. (14) and (15)), this value is very sensitive to those
uncertainties giving rise to relatively large errors.

## 5   Comparison with independent observations

This section deals with two different aspects. In section 5.1 the retrieved radii of stratospheric sulfate
aerosol particles are compared to (not collocated) results from the literature. Section 5.2 presents
comparisons of the retrieved aerosol extinction profiles with available satellite data sets.

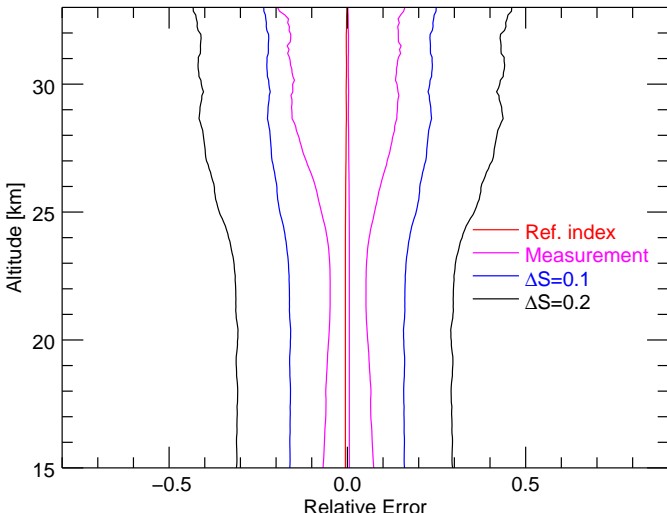

**Fig. 6.** Impact of measurement errors, distribution width and aerosol refractive index on the radius retrieval.



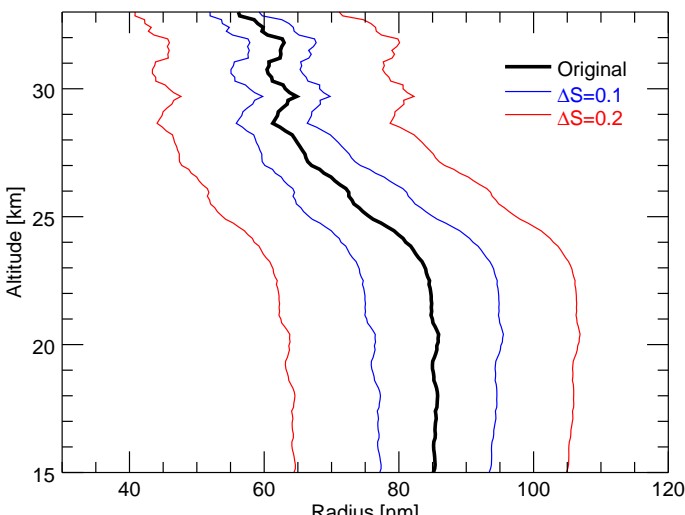

**Fig. 7.** Radius profile inferred with an assumed distribution width of $S = 1.3$ (black curve). The error ranges are obtained by adding all single error contributions (Monthly mean for March 2013).

### 5.1 Comparison of particle size retrievals

The number of available studies on the size of stratospheric sulfate particles is rather limited. Here,
we limit the considerations to qualitative comparisons of non-collocated measurements.
McLinden et al. (1999) employed polarized limb-radiance measurements with the CPFM (Com-
position and Photodissociative Flux Measurement) spectroradiometer operated on NASA's ER-2
high altitude aircraft during two field campaigns in April and May 1997. Both flights started in Fair-
banks, Alaska (65°N, 148°W) and covered the high latitude American sector. For the retrieval of
aerosol particle size information CPFM limb scans at latitudes of 83°N and 75°N, respectively were
used. For their aerosol retrieval McLinden et al. (1999) assumed an altitude independent log-normal
PSD which they claimed it to be representative of the aerosol at all heights in the lower stratosphere
covered by the measurements. Median radii of $r_m = 120 \pm 20$ nm and $100 \pm 20$ nm and logarithmic
standard deviations of $0.44 \pm 0.04$ and $0.46 \pm 0.04$ were retrieved for the two flights. Converted
to the geometric standard deviations, the width values are $S = 1.55 \pm 0.06$ and $S = 1.58 \pm 0.06$, re-
spectively.
Bourassa et al. (2008) retrieved stratospheric aerosol particle size information from OSIRIS limb-
scatter measurements at 750 nm and 1530 nm, also assuming a log-normal PSD. From a single limb-



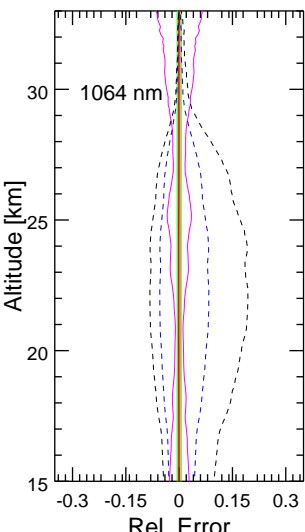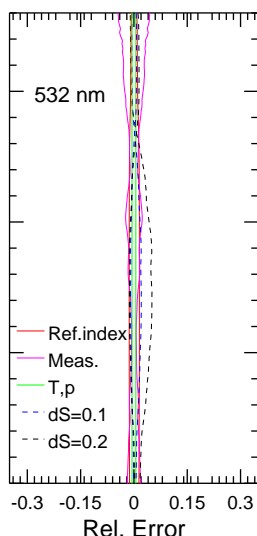

**Fig. 8.** Relative error of the extinction coefficient due to uncertainties in refraction index, temperature, distribution width and measurement error of monthly mean values for March 2013.

scan – assuming a geometric standard deviation of 1.6 – the authors retrieved a *median* (the article
incorrectly states *mode*) radius, decreasing from about 100 nm at 21 km to about 30 nm at 30 km
altitude. The analyzed limb scan was performed on January 5, 2004 at -35.6° S latitude and 112.6°
longitude.
Ugolnikov et al. (2018) presented stratospheric aerosol particle size retrievals from ground-based
multi-spectral twilight measurements carried out with an all-sky camera in central Russia (55.2°N,
37.5°E) in spring and summer 2016, i.e. for volcanically relatively quiescent conditions. The authors
assumed a log-normal PSD retrieved a *mean* aerosol radius of about $r_m = 80$ nm and a width of
$S = 1.5 - 1.6$.
Bingen et al. (2004a); Bingen at al. (2004b) retrieved stratospheric aerosol particle size informa-
tion from SAGE II solar occultation measurements for the period from 1985 to 2000. The retrieved
aerosol radii – it is unclear which radius is shown, because both the terms *mode* and *median* are used
– are on the order of $200 - 350$ nm at an altitude of 22.5 km in 1999, i.e. 8 years after the eruption
of Mt. Pinatubo, when the stratospheric aerosol load was again close to background conditions. It
is, however, important to mention that the retrieved PSDs are rather narrow, with $S \leq 1.1$.
Deshler (2008) retrieved stratospheric aerosol particle size from balloon-borne measurements
done at Laramie (Wyoming, USA, 41°N, 105°W). This analysis considered a bimodal size dis-

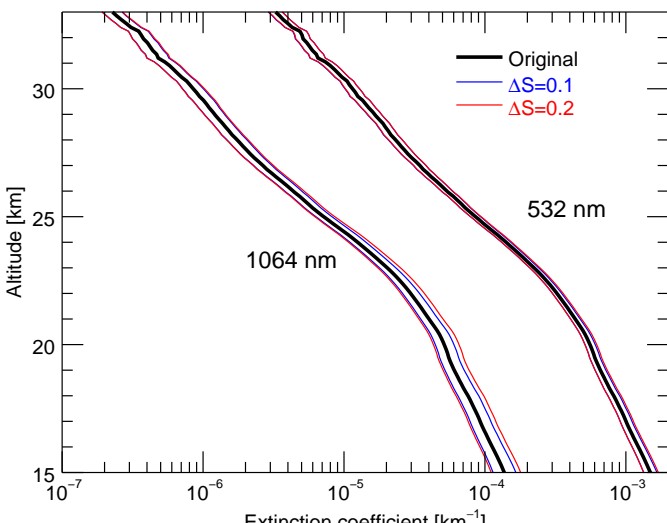

**Fig. 9.** Extinction coefficient profiles inferred with an assumed distribution width of $S = 1.3$ with total error ranges of the two assumed deviations of the distribution width (Monthly mean for March 2013.)

tribution which may be present even at background aerosol conditions. Averaged profiles obtained
between 1995 and 2003 without volcanic aerosol load from Pinatubo show an increase of the median
radius of the main mode from roughly 45 nm at 16 km up to a maximum of 80 nm at 21 km with a
distribution width of S=1.37. For the time period with volcanic aerosols a maximal median radius
of the main mode is found to be around 180 nm with S=1.41.
In summary, a majority of the limited number of studies on the size of stratospheric sulfate aerosols
yields – under volcanically quiescent conditions – median radii on the order of about 100 nm, in
good overall agreement with the retrievals presented here. The exception are size retrievals based
on multi-spectral solar occultation measurements with SAGE II (Bingen et al., 2003, 2004a; Bingen
at al., 2004b) yielding mode (or median radii, this distinction was not made by the authors) of
several hundred nm, even in the late 1990s, when the Pinatubo aerosol has already almost entirely
disappeared. These discrepancies may in part be a consequence of different sensitivities to the
aerosol particle population in combination with errors in the assumed PSD.

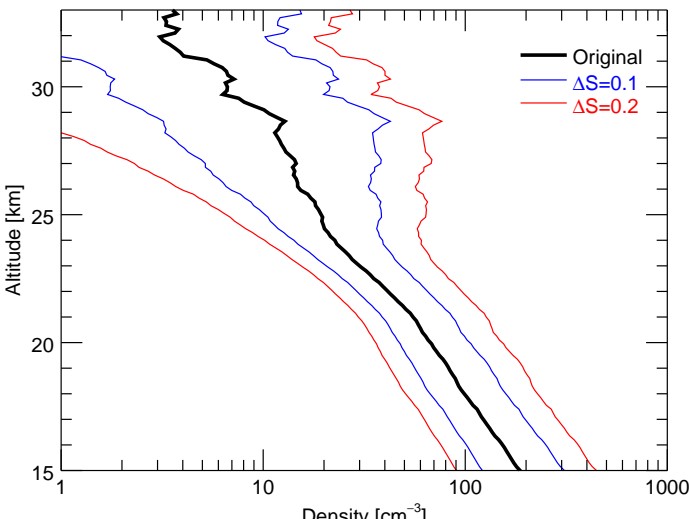

**Fig. 10.** Aerosol density profile inferred with an assumed distribution width of $S = 1.3$ with total error ranges (Monthly mean for March 2013.)

### 5.2 Comparison with satellite aerosol extinction observations

In this section the extinction coefficient profiles retrieved from the measurements with the ALOMAR
lidar are compared to aerosol extinction profiles retrieved from measurements with different satellite
instruments. Specifically, we use observations from two instruments.
OSIRIS (Optical Spectrograph and InfraRed Imaging System) (Llewellyn et al., 2004) was launched
in 2001 on-board the Swedish/Canadian/Finish/French Odin satellite (Murtagh et al., 2002). OSIRIS
performs limb-scatter observations in the $280 - 800\,\mathrm{nm}$ spectral range, allowing to retrieve strato-
spheric aerosol extinction profiles (Bourassa et al., 2012). Aerosol extinction is provided at a wave-
length of 750 nm (data version 5.07).
OMPS (Ozone Mapping Profiling Suite) was launched in 2012 on-board the Suomi-NPP (Suomi
National Polar-orbiting Partnership) satellite and performs nadir and limb-scatter measurements (e.g.
Jarros et al., 2016). Stratospheric aerosol extinction profiles are retrieved from limb-scatter measure-
ments and are provided at a wavelength of 675 nm (data version 1) (DeLand et al., 2016).
The wavelengths, at which the aerosol extinction coefficients are provided, the measurement ge-
ometry and covered time span are summarized in table 2.
For comparison monthly mean and zonally averaged data for March 2013 in the latitude range





| Instrument | Satellite | Time span | Geometry | $\lambda$ [nm] |
|---|---|---|---|---|
| OSIRIS | Odin | 2001-2017 | Limb-scatter | 750 |
| OMPS | Suomi-NPP | 2012-2017 | Limb-scatter | 675 |

**Table 2.** Overview on the satellite data used for comparison.

60°N-80°N was used. Measurements performed in March should not be influenced by polar strato-
spheric clouds which frequently appear during winter months. Prior to the comparison the extinc-
tion profiles measured at the two wavelengths 1064 nm and 532 nm have to be converted to the
wavelengths for which the satellite data is provided. This conversion is done using the Ångström
approach (Ångström, 1929)

$$k_i(\lambda_i, z) = c(z) \cdot \lambda_i^{-\alpha(z)} \tag{16}$$

with the Ångström exponent $\alpha(z)$ and $\lambda_i$ denoting the considered or desired wavelength. Since
our retrieval provides extinction coefficients profiles at two wavelengths the Ångström exponent is
obtained by

$$\alpha(z) = \frac{\ln(k_{ext}(z, \lambda_2)) - \ln(k_{ext}(z, \lambda_1))}{\ln(\lambda_1) - \ln(\lambda_2)}. \tag{17}$$

A comparison between the satellite observations and the lidar extinction profiles converted to the
corresponding satellite wavelength is shown in Figure 11.

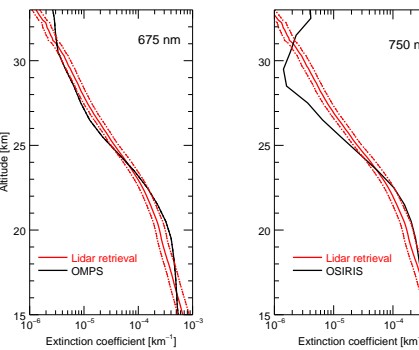

**Fig. 11.** Comparison of the re-scaled lidar extinction profiles with a maximal error range taken from Figure 9
(S=1.3) with profiles obtained from the OSIRIS and OMPS instruments for March 2013.

Relative deviations between the lidar extinction coefficient scaled to the wavelength used by the
satellite and the satellite measurement are presented in Figure 12. The shapes of both profiles look
very similar for altitudes below roughly 25 km. Above this altitude however, the OSIRIS profile



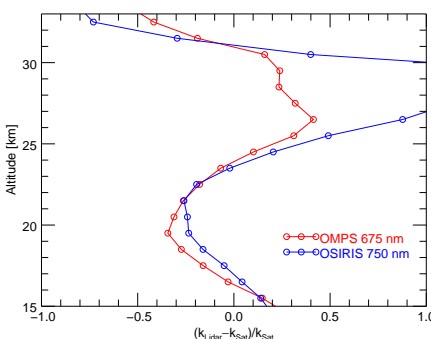

**Fig. 12.** Relative difference between the lidar extinction profile scaled to the satellite wavelength and satellite measurements for March 2013. The lidar extinction profiles were retrieved assuming with S=1.3.

shows a pronounced dip-like structure. This behaviour is probably caused by a sporadically appear-
ing low bias at high latitudes and altitudes of the device, which is a known issue (Rieger et al, 2015).
It is worth pointing out that the measurements are not performed at exactly the same location.

## 6  Conclusions

In this work we present an approach for retrieving particle size and extinction coefficient profiles
of the stratospheric aerosol layer from multi-color measurements with the ALOMAR-RMR lidar
in northern Norway. The retrieval approach is based on comparing measured and modelled color
ratios of the wavelengths 1064 nm and 532 nm. In a first retrieval step profiles of the aerosol median
radius – assuming a log-normal particle size distribution with fixed width – are obtained. These
are used in a second step – together with temperature and density profiles – to calculate the desired
aerosol extinction profiles. Although assumptions on the aerosol properties have to be made, the
inferred extinction coefficients are relatively robust against variations of the assumed distribution
width, which constitutes the dominant error source. Additionally, aerosol density profiles were
computed, though with larger uncertainties.
The median radii of the aerosol size distribution obtained with this approach are in good overall
agreement with other independent particle size measurements (except those by SAGE II) which
confirm that our assumptions are valid. The consequence of a potential low bias in aerosol size is a
high bias in aerosol density. In the case of density profiles, however, the broad error ranges should
be kept in mind.
Finally, the direct comparison with extinction coefficient profiles obtained by satellite-borne mea-
surements show a significant relative difference of roughly 40% at altitudes around 20 km which
in case of OSIRIS can reach over 100% at higher altitudes. As pointed out, this high deviation is



probably caused by a low bias of the aerosol extinction retrieved from OSIRIS measurements at high
northern latitudes and the respective altitudes.

## Code and data availability

The datasets used in this study can be obtained by contacting the first author.

## Competing interests

The authors declare that they have no conflict of interest.

## Author contribution

Christian von Savigny and Jacob Zalach developed and implemented the described method whith
the help of other authors.
*Acknowledgements.* This work was funded by the Deutsche Forschungsgemeinschaft (DFG, grant SA-1351/7)
and supported by University of Greifswald as well as the Leibniz Institute of Atmospheric Physics at Kühlungsborn.



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
