# Peer review of "Challenges in retrieving stratospheric aerosol extinction and particle size from ground-based RMR-LIDAR observations"

_Atmospheric Measurement Techniques, 2019_

## Referee Comment (RC1) · Anonymous Referee #1 · 5 Sep 2019

The paper is to my opinion not ready for publication. The authors present a lidar retrieval technique which is not new and thus does not justify publication. The presented sensitivity study belongs to the description of the methodology Thus, the sensitivity study also does not justify publication. However, a new aspect would be if the authors would apply their (updated) lidar method to observations from 2000 to 2019 (they have these observations, this is mentioned in the abstract).

Therefore, presently, I recommend rejection. I would accept the manuscript if they concentrate on the data analysis and show long-term measurement results and discuss them. At least if there would be a section with new results (2000-2019 observations).

[Figure]

Details

Abstract: What aerosol type do you expect in the stratosphere? Nothing is specified. Obviously only aged volcanic sulphuric acid solution droplets (so that Mie scattering in fully applicable). But, volcanic ash or biomass burning smoke (injected by pyrocumulonimbus activity) can be present as well. These particles may be non-spherical and may have quit complex refractive index characteristics.

Section 2.

P3, L61: The RMR lidar is described in von Zahn et al. (2000). . ... My question: No new aspects, new channels, new receiver design etc. . . since then, no published updates (articles)?

Obviously the RMR lidar has no depolarization channels, and thus no information about particle shape and thus aerosol type is available.

P4, L95: The sulphuric acid content (here of 75%) is not fixed and changes with temperature.

P4, L101-106: What about papers of Jaeger et al., Hofmann et al., Deshler et al. (mostly in GRL and JGR)? They use balloon (in situ) observations of aerosol size distributions over Wyoming and published microphysical properties of measured stratospheric aerosols . . . Are these papers (and measurements) in agreement with your assumptions?

P5, Eq.(5), why is $k\_Mie$ in the equation and not $k\_Ray$?

$k\_Mie/Ray$ , $P\_Mie/Ray$ is misleading, could be related to a Mie/Ray signal ratio.

P5, L121-125: The profile of the particle backscatter coefficient at 1064 nm is not easy to calibrate. How do you do that in practice. Use of cirrus? Assuming same backscatter coefficient at 532 and 1064 nm? Please explain.

So, at the end of subsection 3.3 I already asked myself: What is new?

P7: You computed the monthly mean backscatter coefficient profiles first. Then you calculated color ratio, median radius, etc.! Does that make sense? I was expecting, you make use of single nighttime observations. For one night, the aerosol characteristics may be constant, but over a whole month (30 days)?

Fig.2: Shown is NOT the monthly mean Median Radius! Shown is the Median Radius computed from the monthly mean backscatter coefficient.

Again, in Fig 3, the extinction coefficient is derived from the monthly mean data of aerosol backscatter profiles . . . and not obtained from individual night-by-night observations and subsequent averaging!

The lidar ratio (Eq. 12) is the solution of an ill-posed problem and thus not just reliable.

At the end of section 4, I asked myself again: What is new?

The presented comparisons in section 5 are not very convincing!

So, my conclusion is: One should analyse the available 2000-2019 stratospheric lidar data set. If thes key findings are added, then a publication may be justified (provided the presentations, discussions, and conclusions are acceptable).

Without these observations, I recommend: rejection.

———————————————————

---

## Author Comment (AC1) · 12 Sep 2019

With this comment we would like to respond to the comment published by anonymous reviewer #1 on 05 September 2019.

We are sorry that the essential aspects of our work were not clear to the reviewer. So we would like to clarify these. The described methodology may be a standard approach for tropospheric aerosol studies, but it has essentially never been applied to study stratospheric aerosols. The novelty of the work is that the method is applied to stratospheric aerosols. And given (a) the large differences in stratospheric aerosol size published in earlier studies and (b) the overall importance of stratospheric sul-

fate aerosols (particularly of volcanic origin) for climate, we strongly believe that the manuscript is of relevance for the community and worth to be published. We think that the reviewer's main argument (nothing new) is not applicable to our manuscript.

Furthermore, most studies on lidar measurements of stratospheric aerosols assume a value of the lidar ratio, i.e. implicitly assume specific values of the particle size distribution. This is not necessary with the method presented in our paper, because the particle size is retrieved in a first step, followed by the retrieval of aerosol extinction.

We also would like to emphasize, that the described methodology can be applied to measurements with any Rayleigh/Mie/Raman lidar with sufficient data quality and measurements at 387, 608, 532, 1064 nm.

We will respond to the specific points raised by the reviewer in the near future.

With kind regards,

Jacob Zalach on behalf of all authors
* * *

---

## Referee Comment (RC2) · Anonymous Referee #2 · 20 Sep 2019

Here is a paper describing a method to analyze monthly mean lidar color ratios to infer: profiles of the median radius of lognormal size distributions which model the measurements, the lidar ratio, and finally extinction from backscatter measurements. Yet no examples of the fundamental data are shown.

The paper moves quickly, and with no justification, from distributions of quantities with respect to color ratio, the measurement, to distributions of derived quantities with altitude. There is no explanation of how this transformation is made, yet the results from the rest of the paper hinge on this.

The method to derive lidar ratio merits almost no explanation, yet it is a method I have

never seen before, and raises questions as to why others have not used this method.

There are errors in some of the equations, and the origin of equation (11) is unclear. I detail these comments and questions in the review below.

10 Budget

17-32 This nice description of the importance of stratospheric aerosol would benefit from some additional appropriate references.

Eq (5) Shouldn't the scattering term k be k(sca)_Ray?

116 For consistency with the ratio on k(sca) on line 115 change to "depend on aerosol/air densities. I don't believe that there is any Mie scattering from air molecules.

118 The scattering cross section, $\sigma$, should also be defined here, or above.

130 Why is n0($\lambda$) given only for 532 nm? What about 1064 nm?

142 Probably should add a more standard reference for Mie scattering (e.g. Born and Huffman or Dave), to which the Oxford scattering calculations surely have probably been compared. Perhaps this is even referenced in their code.

177-178 Why do monthly mean data rule out a distribution width of 1.1? If the aerosol signal was from uniformly narrow distributions over the month then this is possible. Background stratospheric aerosol is thought to be from generally a rather well constrained and somewhat stable size distribution. The authors need a better argument to rule out a width of 1.1. The authors could look to the literature. It should be quite straight forward to find a reference to a typical background aerosol size distribution, for example from in situ measurements, which would not be consistent with a narrow size distribution with a median radius near 300 nm. This would clearly rule out a width of 1.1.

180-181 characterise should be characterize.

Figure 2 Why is the ordinate altitude and not color ratio as Figure 1. The factors involved in this transformation of the ordinate are not clear. In any case it seems that for branch one the radius variation range is limited to about 20 nm once the distribution width is assumed. This is quite restrictive.

183-185 For the reader to understand this statement they would have to know how the color ratio varies with altitude. Isn't it enough for both branches 2 and 3 to point out that in these cases approximately half of the color ratio range would not be covered?

190 Change 'reduces' to 'restricts'. It's easy to misinterpret the sentence, as I did, if reduces is used, to mean the application to radii < 150 nm is limited.

196 '... eqs (3, 4, and 5) can be solved for ...'

Figures 2, 3, 4. There is some important information missing which is required to allow the reader to understand and tie Figures 2-4 to Figure 1. That information is the vertical distribution of the color ratio and for figures 3 and 4 the vertical distribution of the scattering ratio. After Figure 1 the ordinate shifts from color ratio to altitude with no explanation of how the two are related. In Figure 1 the color ratio range is 0.1-3.8. So how is this color ratio distributed by altitude? Once this is known then maybe it will be clear how the following figures are generated.

The two altitude dependent quantities in Eq (11) are P_Mie through its dependence on r_m and the scattering ratio, R. But for the lidar ratio the authors claim that only P_Mie is required and the altitude dependence is through r_m and hence the color ratio. All the vertical profiles, except the lidar ratio at 1064 nm, decrease rather significantly at 23 km, right where r_m decreases from 80 nm towards 60 nm for s=1.3. Is this all that's driving this vertical structure? And if that is the case is the lidar ratio at 532 nm really that much more sensitive to a change in radius from 80 to 60 nm than the lidar ratio at 1064 nm. There should be more discussion on these points.

Eq (11) How is this Equation used? The term k_Ray * P_Ray($\pi$) in the numerator on

the RHS of Eq (11) is calculated from the molecular density profiles from ERA-interim, and this same term, handled the same way, appears in the denominator of (R-1). So if Eq (11) is simplified it is a simple statement that k_Mie = k_Mie*P_Mie($\pi$)/P_Mie($\pi$) or k_Mie=k_Mie. Isn't this a tautology? So how is Eq (11) something more than the measured backscatter divided by the phase function for backscatter, which can be calculated once the particle size is assumed and the wavelength known?

205-213 There has been a lot of previous work devoted to determining the lidar ratio, but I have not seen the approach here. Is it really as simple as inverting the backscatter phase function, with the assumption that the backscatter is just the scattering coefficient times the phase function? Don't equations 12 and 13 imply that P_Mie (0) is 1? Earlier, line 114, k_Mie/Ray were defined as scattering coefficients, now here that term is being equated to extinction, the sum of scattering and absorption. Also if the lidar ratio is just the inverse of P_Mie($\pi$,r_m,$\lambda$), why isn't that method used by, for example Jaeger et al. (1995), to calculate the lidar ratio from measured size distributions?

217 Or for certain wavelengths, 1064 nm?

230 They are identical because they both use a calculated cross section integrated over the same size distribution, or? It's hard to believe they would be identical if they were derived from measurements. But so far we haven't seen any measurements.

242-245 Why are errors in temperature and pressure stated if they are not required because they cancel out? This is just a waste of the reader's time.

Figure 7 How is this figure different from an expanded version of Figure 2? It is basically the same figure. What are all the error contributions included? It was already stated that temperature and pressure cancel each other out because color ratios are used, so these are not included. It is not explained how a difference in the refractive index affects the retrieved radius.

Figure 8 What kind of a scale is on the abscissa. The minor tick marks cannot be used

to state what the precision is exactly, but it appears to be less than 3% for 532 nm. This figure then indicates that a difference in aerosol radius between the min and max distribution width, which is about a factor of two in radius, lead to almost no effect on extinction? This is a surprising result, suggesting that the determination of the median radius is not that critical. There is a much larger effect at 1064 nm but it seems a bit odd that the effect is not symmetric.

254-258 According to Eq (11) the only size distribution information used is the median radius, $r\_m$, which appears in the phase function. Thus I don't follow this argument that the uncertainties in distribution width are compensated for by the opposite uncertainty in median radius. The uncertainty in distribution width leads to the radius uncertainty which is then used in calculating extinction, according to Eq (11).

Figure 9 What is the point of this figure. It is just a repeat of Figure 3 with the per cent uncertainties, already shown in Figure 8, added to absolute extinction, and it is much less helpful than Figure 8 in assessing this uncertainty.

Eq (16) c(z) is not defined. Shouldn't there be a ratio of wavelengths within the exponential term of the desired wavelength over the reference wavelength.

Figure 11 Which lidar extinction profile is used in the Angstrom conversion to the satellite wavelengths.

Jäger, H., T. Deshler, and D. J. Hofmann, Midlatitude lidar backscatter conversions based on balloonborne aerosol measurements, Geophys. Res. Lett., 22, 1729-1732, 1995.

---

## Author Comment (AC2) · 13 Dec 2019

We thank the reviewer for his comments and suggestions. In addition to our general answer from 12. September we would like to respond to the details of the comment published by anonymous reviewer #1 on 05 September 2019.

Referee comment: "The paper is to my opinion not ready for publication. The authors present a lidar retrieval technique which is not new and thus does not justify publication. The presented sensitivity study belongs to the description of the methodology Thus, the sensitivity study also does not justify publication.

However, a new aspect would be if the authors would apply their (updated) lidar method to observations from 2000 to 2019 (they have these observations, this is mentioned in the abstract). Therefore, presently, I recommend rejection. I would accept the manuscript if they concentrate on the data analysis and show long-term measurement results and discuss them. At least if there would be a section with new results (2000-2019 observations)."

Our response: We are sorry that the essential point of our work was hard to find in the text and we hope to have clarified it with our first answer. As a short reminder we can state that the novel aspect lies in the methodology itself which - to our best knowledge - has never been applied to study stratospheric aerosols. One feature of our approach is, in contrast to most studies of lidar measurements of stratospheric aerosols, that we do not need do assume a value of the lidar ratio.

Taking into account the large differences in stratospheric aerosol size published in earlier studies and the overall importance of stratospheric sulphate aerosols (particularly of volcanic origin) for climate, we strongly believe that the manuscript is of relevance for the community and worth to be published.

Based on this we would strongly prefer to separate the publication of the method itself and the results obtained with it.

Referee comment: "Abstract: What aerosol type do you expect in the stratosphere? Nothing is specified. Obviously only aged volcanic sulphuric acid solution droplets (so that Mie scattering in fully applicable). But, volcanic ash or biomass burning smoke (injected by pyrocumulonimbus activity) can be present as well. These particles may be non-spherical and may have quit complex refractive index characteristics."

Our response: We expect stratospheric aerosols with properties listed in section 3.1. Essentially we expect a spherical droplet of water and sulphuric acid without absorption and a log-normal size distribution. All other potential types of stratospheric aerosols are neglected in this study. Non-sulphate aerosols are planned to be considered in

future work. We also note that many studies on stratospheric aerosols assume that the stratospheric aerosol population is strongly dominated by suphate aerosols (Bingen at al (2004a), Bingen at al (2004b), McLinden et al. (1999) , Bourassa et al. (2008) , Taha et al. (2011), von Savigny et al (2015)).

Referee comment: "Section 2. P3, L61: The RMR lidar is described in von Zahn et al. (2000). . .. My question: No new aspects, new channels, new receiver design etc. . . since then, no published updates (articles)? Obviously the RMR lidar has no depolarization channels, and thus no information about particle shape and thus aerosol type is available."

Our response: As we hope to made clear, the novelty is the application of this method to stratospheric aerosols. Since the article focuses on the description of the method, the lidar device itself is not described here, we just reference the main work concerning it (Langenbach at al (2019) and von Zahn et al (2000))

Referee comment: "P4, L95: The sulphuric acid content (here of 75%) is not fixed and changes with temperature. "

Our response: This is true but this assumption is made very often. We assume that the change of the sulphuric acid content is considered through its influence on the refractive index, which we vary in section 4 in order to quantify its influence on the results. Our variation of the refractive index corresponds to a sulphuric acid concentration variation of 20%.

Referee comment: "P4, L101-106: What about papers of Jaeger et al., Hofmann et al., Deshler et al. (mostly in GRL and JGR)? They use balloon (in situ) observations of aerosol size dis- tributions over Wyoming and published microphysical properties of measured strato- spheric aerosols . . . Are these papers (and measurements) in agreement with your assumptions?"

Our response: In section 5.1 we compare our results with works of McLinden et al.

(1999), Bourassa et al. (2008), Ugolnikov et al. (2018), Bingen et al. (2004) and Deshler (2008).

Under volcanically quiescent conditions most of this studies are in good overall agreement with our results. Only the work of Bingen et al. (2003, 2004a, 2004b) which analyses SAGE II data yields much larger radii of several hundred nm, even in the late 1990s, when the Pinatubo aerosol has already almost entirely disappeared. These discrepancies may in part be a consequence of different sensitivities to the aerosol particle population in combination with errors in the assumed PSD.

Referee comment: "P5, Eq.(5), why is k_Mie in the equation and not k_Ray?"

Our response: Thank you, this was obviously wrong.

Referee comment: "k_Mie/Ray , P_Mie/Ray is misleading, could be related to a Mie/Ray signal ratio."

Our response: Thank you for pointing out this misleading expression. We have corrected this.

Referee comment: "P5, L121-125: The profile of the particle backscatter coefficient at 1064 nm is not easy to calibrate. How do you do that in practice. Use of cirrus? Assuming same backscatter coefficient at 532 and 1064 nm? Please explain. "

Our response: Please note that we do not make use of the particle backscatter coefficient at 1064 nm to calculate the color ratio from lidar data, we just use the backscatter ratios (First line of Equation 8). The following discussion is to show how the colour ratio depends on the Mie backscatter coefficients and how to calculate the colour ratio for the forward model.

The backscatter coefficient at 1064 nm is calculated from the backscatter ratio, the ECMWF model density and the Rayleigh backscatter coefficient $\beta_{Mie}(z, 1064 nm) = \beta_{Ray}(z, 1064 nm) *(R(z, 1064nm)-1)$. In our case the we normalize the backscatter coefficient at 1064 nm to the Rayleigh backscatter coefficient at 34 to 38

km altitude (See Langenbach et al., (2019) for the normalization procedure).

Referee comment: "So, at the end of subsection 3.3 I already asked myself: What is new?"

Our response: We have tried to make the new aspects more obvious in the revised manuscript. One core aspect is the simplicity of the method. It produces results which compare well to satellite data under the current aerosol load of the stratosphere.

Referee comment: "P7: You computed the monthly mean backscatter coefficient profiles first. Then you calculated color ratio, median radius, etc.! Does that make sense? I was expecting, you make use of single nighttime observations. For one night, the aerosol characteristics may be constant, but over a whole month (30 days)?"

Our response: Your objection is of course right, but in this work we want to focus on the description of the method. The detailed analysis of single night-time measurements with a high sample rate can of course be done and will be made in the future. But such an analysis would require a discussion of the results which would take the focus away from the method applied. We wanted to avoid such a situation.

Referee comment: "Fig.2: Shown is NOT the monthly mean Median Radius! Shown is the Median Radius computed from the monthly mean backscatter coefficient Again, in Fig 3, the extinction coefficient is derived from the monthly mean data of aerosol backscatter profiles . . . and not obtained from individual night-by-night observations and subsequent averaging!"

Our response: The reviewer is absolutely right, our description was not correct. Thank you for pointing this out. The sentences were adjusted accordingly.

Referee comment: "The lidar ratio (Eq. 12) is the solution of an ill-posed problem and thus not just reliable. At the end of section 4, I asked myself again: What is new?"

Our response: As hopefully clarified the novelty of our method is its application on stratospheric aerosols. Concerning Eq.12 we do not claim that this approach is reliable,

but it is often applied and here we compare this approach with our method. We would appreciate if you could be more specific here.

Referee comment: "The presented comparisons in section 5 are not very convincing!"

Our response: Our comparison includes all relevant aspects - but you are right, the agreement is not that good. However, we would like to point out that data sets provided by different satellite measurements also often deviate from each other.

Referee comment: "So, my conclusion is: One should analyse the available 2000-2019 stratospheric lidar data set. If thes key findings are added, then a publication may be justified (provided the presentations, discussions, and conclusions are acceptable). Without these observations, I recommend: rejection."

Our response: At this point we would like to express our hope again that we were able to better show the essential point of our work, the application of the described method on stratospheric aerosols. Therefore, we would strongly prefer to publish the obtained results in an future, separate article.

Literature used:

Bingen, C., Fussen, D., and Vanhellemont, F.: A global climatology of stratospheric aerosol size distribution parameters derived from SAGE II data over the period 1984-2000: 1. Methodology and climatological observations, J. Geophys. Res., 109, D06201, doi:10.1029/2003JD003518, 2004a

Bingen C., Fussen D., and Vanhellemont F.: A global climatology of stratospheric aerosol size distribution parameters derived from SAGE II data over the period 19842000: 2. Reference data, J. Geophys. Res., 109, D06202, doi:10.1029/2003JD003511, 2004b

Bourassa, A. E., Degenstein, D. A., and Llewellyn, E. J.: Retrieval of stratospheric aerosol size information from OSIRIS limb scattered sunlight spectra, Atmos. Chem. Phys., 8, 6375-6380, 2008

Deshler, T.: A review of global stratospheric aerosol: Measurement, importance, life cycle, and local stratospheric aerosol, Atmos. Res., 90, 223-232, doi:10.1016/j .atmosres.2008.03.016, 2008

McLinden, C. A., McConnell, J. C., McElroy, C. T., and Grif?oen, E.: Observations of stratospheric aerosol using CPFM polarized limb radiances, J. Atmos. Sci., 56, 233-240, 1999

Taha, G., Rault, D. F., Loughman, R. P., Bourassa, A. E., and von Savigny, C.: SCIA-MACHY stratospheric aerosol extinction profile retrieval using the OMPS/LP algorithm, Atmos. Meas. Tech., 4, 547-556, doi:10.5194/amt-4-547-2011, 2011.

Ugolnikov, O. S. and Maslov, I. A.: Investigations of the Background Stratospheric Aerosol Using Multi- color Wide-Angle Measurements of the Twilight Glow Background, Cosmic Research, 56, 2, 85-93, doi:10.1134/S0010952518020119, 2018, Original Russian Text, 2018, published in Kosmicheskie Issledovaniya, 2018, 56, 2, 95-102, 2018

von Savigny, C., Ernst, F., Rozanov, A., Hommel, R., Eichmann, K.-U., Rozanov, V., Burrows, J. P., and Thomason, L. W.: Improved stratospheric aerosol extinction profiles from SCIAMACHY: validation and sample results, Atmos. Meas. Tech., 8, 5223 - 5235, 2015.

Please also note the supplement to this comment: https://www.atmos-meas-tech-discuss.net/amt-2019-267/amt-2019-267-AC2-supplement.pdf
* * *

---

## Author Comment (AC3) · 13 Dec 2019

With this document we would like to respond to the comment published by anonymous reviewer #2 on 20 September 2019. We thank the reviewer for his helpful comments, which were essentially all considered in the revised version of the manuscript.

Referee comment: "Here is a paper describing a method to analyze monthly mean lidar color ratios to infer: profiles of the median radius of lognormal size distributions which model the measurements, the lidar ratio, and finally extinction from backscatter measurements. Yet no examples of the fundamental data are shown."

[Figure]

Our response: We changed this and included now an example of the measured data. In section 2 we have included a new Figure 1 showing measured backscatter ratio profiles for 532 and 1064 nm with error bands together with the corresponding color ratio.

Referee comment: "The paper moves quickly, and with no justification from distributions of quantities with respect to color ratio, the measurement, to distributions of derived quantities with altitude. There is no explanation of how this transformation is made, yet the results from the rest of the paper hinge on this."

Our response: The general concept of our method is described in section 3. We added text with further expanations to section 3.2 and hope this point is now clearer. Especially the origin of the altitude scale should now be better understandable.

Referee comment: "The method to derive lidar ratio merits almost no explanation, yet it is a method I have never seen before, and raises questions as to why others have not used this method."

Our response: We are sorry, but we do not fully understand this comment. Our paper is wholly dedicated to explain the method, so we would appreciate if you could be more specific on this point. If the particle size distribution is known (or has been estimated), then the determination of the lidar ratio is indeed that simple. However, in many cases, this information is not available.

Referee comment: "There are errors in some of the equations, and the origin of equation (11) is unclear. I detail these comments and questions in the review below. 10 Budget"

Our response: Thank you, somehow the spell checking has overseen this.

Referee comment: "17-32 This nice description of the importance of stratospheric aerosol would benefit from some additional appropriate references."

Our response: Now we included more references as suggested.

Referee comment: "Eq (5) Shouldn't the scattering term k be k(sca)_Ray?"

Our response: Thank you, this was obviously wrong.

Referee comment: "116 For consistency with the ratio on k(sca) on line 115 change to "depend on aerosol/air densities. I don't believe that there is any Mie scattering from air molecules."

Our response: We changed it.

Referee comment: "118 The scattering cross section, $\sigma$, should also be defined here, or above."

Our response: The $\sigma$ was not mentioned in the text. We have improved this and provided a reference.

Referee comment: "130 Why is n0($\lambda$) given only for 532 nm? What about 1064 nm?"

Our response: Now the value for 1064 nm is also given.

Referee comment: "142 Probably should add a more standard reference for Mie scattering (e.g. Born and Huffman or Dave), to which the Oxford scattering calculations surely have probably been compared. Perhaps this is even referenced in their code."

Our response: In the source code of the software package used (Mie scattering routines (2018)) is a short reference to Bohren and Huffmann (1998). We have added this reference to the text.

Referee comment: "177-178 Why do monthly mean data rule out a distribution width of 1.1? If the aerosol signal was from uniformly narrow distributions over the month then this is possible. Background stratospheric aerosol is thought to be from generally a rather well constrained and somewhat stable size distribution. The authors need a better argument to rule out a width of 1.1. "

Our response: The work of Langenbach et al (2019) shows that even at high altitudes

between 23 and 32 km and on short time scales of several hours the stratospheric background aerosol layer is highly dynamic. Therefore, the assumption of an aerosol population with a very narrow distribution width during a relatively long time period of one month is at least problematic. We included this in the text in section 3.3

Referee comment: "The authors could look to the literature. It should be quite straight forward to find a reference to a typical background aerosol size distribution, for example from in situ measurements, which would not be consistent with a narrow size distribution with a median radius near 300 nm. This would clearly rule out a width of 1.1."

Our response: That's right. In section 5.1 we compare our results with works of McLinden et al. (1999), Bourassa et al. (2008), Ugolnikov et al. (2018), Bingen et al. (2004) and Deshler (2008).

Under volcanically quiescent conditions most of this studies are in good overall agreement with our results. Only the work of Bingen et al. (2003, 2004a, 2004b) which analyses SAGE II data yields much larger radii of several hundred nm, even in the late 1990s, when the Pinatubo aerosol has already almost entirely disappeared. These discrepancies may in part be a consequence of different sensitivities to the aerosol particle population in combination with errors in the assumed PSD.

The available studies providing experimental values on sigma show sigmas exceeding a value of 1.1. A direct reference to other works which derived aerosol distribution widths with values above S=1.4 is given in section 3.3, lines 175-178 (McLinden et al., 1999; Bourassa et al., 2008; Ugolnikov et al., 2018).

Referee comment: "180-181 characterise should be characterize."

Our response: Thank you, now it is corrected.

Referee comment: "Figure 2 Why is the ordinate altitude and not color ratio as Figure 1. The factors involved in this transformation of the ordinate are not clear. "

Our response: The transformation of the ordinate axis from colour ratio to altitude is now described in the text in section 3.2.

Referee comment: "In any case it seems that for branch one the radius variation range is limited to about 20 nm once the distribution width is assumed. This is quite restrictive."

Our response: The relatively small radius range retrieved using this approach is not directly restricted by the approach itself, but by the measurements used as an example in Figure 2. For S=1.3, e.g., particle radii from over a range covering about 100 nm are in principle accessible (see blue line in Figure 2).

Referee comment: "183-185 For the reader to understand this statement they would have to know how the color ratio varies with altitude. Isn't it enough for both branches 2 and 3 to point out that in these cases approximately half of the color ratio range would not be covered?"

Our response: As stated above we have now included example data in section 2 with a new Figure which show measured backscatter ratio profiles for both wavelengths and the corresponding color ratio in dependence on the altitude. We hope this improves the understanding.

Referee comment: "190 Change 'reduces' to 'restricts'. It's easy to misinterpret the sentence, as I did, if reduces is used, to mean the application to radii < 150 nm is limited."

Our response: We have changed this.

Referee comment: "196 '. . . eqs (3, 4, and 5) can be solved for . . .'"

Our response: We have changed this.

Referee comment: "Figures 2, 3, 4. There is some important information missing which is required to allow the reader to understand and tie Figures 2-4 to Figure 1.

That information is the vertical distribution of the color ratio and for figures 3 and 4 the vertical distribution of the scattering ratio. After Figure 1 the ordinate shifts from color ratio to altitude with no explanation of how the two are related. In Figure 1 the color ratio range is 0.1-3.8. So how is this color ratio distributed by altitude? Once this is known then maybe it will be clear how the following figures are generated."

Our response: Thank you for pointing out the importance of the transition from colour ratio to altitude. As suggested we have added text to section 3.2 and included a new Figure with example data showing measured backscatter ratio for both wavelengths and the corresponding color ratio. We hope this helps to better understand this point.

Referee comment: "The two altitude dependent quantities in Eq (11) are P_Mie through its dependence on r_m and the scattering ratio, R. But for the lidar ratio the authors claim that only P_Mie is required and the altitude dependence is through r_m and hence the color ratio. All the vertical profiles except the lidar ratio at 1064 nm, decrease rather significantly at 23 km, right where r_m decreases from 80 nm towards 60 nm for s=1.3. Is this all that's driving this vertical structure? And if that is the case is the lidar ratio at 532 nm really that much more sensitive to a change in radius from 80 to 60 nm than the lidar ratio at 1064 nm. There should be more discussion on these points."

Our response: The Rayleigh extinction coefficient k_ray^sca is also altitude dependent and decreases exponentially with the atmospheric scale height. For longer wavelengths the phase function for backscattering exhibits a weaker dependence on radius because with growing wavelength the size parameter approaches the Rayleigh limit. So, here an dissimilar behaviour can be expected. We added this explanation to section 3.4.

Referee comment: "Eq (11) How is this Equation used? The term k_Ray * P_Ray($\pi$) in the numerator on the RHS of Eq (11) is calculated from the molecular density profiles from ERA-interim, and this same term, handled the same way, appears in the denominator of (R-1). So if Eq (11) is simplified it is a simple statement that k_Mie

= k_Mie*P_Mie($\pi$)/P_Mie($\pi$) or k_Mie=k_Mie. Isn't this a tautology? So how is Eq (11) something more than the measured backscatter divided by the phase function for backscatter, which can be calculated once the particle size is assumed and the wavelength known?"

Our response: There would be an tautology if only theoretical values would be used, but this is not the case here. The essential point is, that with R real measured data enters the equation which contains information about the aerosol.

Referee comment: "205-213 There has been a lot of previous work devoted to determining the lidar ratio, but I have not seen the approach here. Is it really as simple as inverting the backscatter phase function, with the assumption that the backscatter is just the scattering coefficient times the phase function? Don't equations 12 and 13 imply that P_Mie (0) is 1?"

Our response: It is that simple if the particle size distribution is known (or has been estimated). However, in many cases, this information is not available. Concerning equations 12 and 13 it should be kept in mind that in equation 13 with beta_mie measured data is used which allows for computation of the scattering coefficient.

Referee comment: "Earlier, line 114, k_Mie/Ray were defined as scattering coefficients now here that term is being equated to extinction, the sum of scattering and absorption. "

Our response: In section 3.1 we write that aerosol absorption has only an negligible effect and therefore we set it to zero. We added a note that in such cases the scattering and extinction coefficients are the same.

Referee comment: "Also if the lidar ratio is just the inverse of P_Mie($\pi$,r_m,$\lambda$), why isn't that method used by, for example Jaeger et al. (1995), to calculate the lidar ratio from measured size distributions?"

Our response: Jaeger et al. (1995) is not able to compute the lidar ratio based on his

data because it consists of measurements at only one wavelength. Therefore, there is no information about particle size present and the lidar ratio is unknown. He uses a different approach with particle counter measurements with which he computes a lidar ratio and with this the extinction coefficient.

Referee comment: "217 Or for certain wavelengths, 1064 nm?"

Our response: You are right, it is a good approximation for 1064 nm over the whole altitude range but for 532 nm only for a constricted range between 15-23 km. We have adjusted the text accordingly.

Referee comment: "230 They are identical because they both use a calculated cross section integrated over the same size distribution, or? It's hard to believe they would be identical if they were derived from measurements. But so far we haven't seen any measurements."

Our response: The profiles are derived from real measured data as stated above concerning the meaning of equation 11, an example of measured data is now included as suggested. They are identical, essentially because measurements at both wavelengths are used to determine the colour ratio and finally the particle size.

Referee comment: "242-245 Why are errors in temperature and pressure stated if they are not required because they cancel out? This is just a waste of the reader's time."

Our response: The influence of temperature and pressure cancels out only for the radius computation. Both values enter the retrieval when the extinction coefficient is derived, they are implicitly included in the value of k_ray. We have added this information into the text.

Referee comment: "Figure 7 How is this figure different from an expanded version of Figure 2? It is basically the same figure. What are all the error contributions included? It was already stated that temperature and pressure cancel each other out because color ratios are used, so these are not included. It is not explained how a difference in

the refractive index affects the retrieved radius."

Our response: Figure 2 shows radius profiles obtained for different assumed distribution widths whereas Figure 7 shows the influence of parameter variation on a radius profile when a distribution width is already chosen, in our case it is S=1.3.

Figure 7 is embedded in section 4, which deals with the error estimation. The assumptions made for the estimation of errors are clearly stated in this section in our opinion and reference to Figure 7 is made. We apologize, if this is not fully clear. The content of the error contribution is specified in lines 249-251: "If the single error contributions are simply added to a total error separately for the two assumed deviations of the distribution width an absolute error range can be assigned to the retrieved radius profile as shown in Figure 7. " This also includes the influence of the refractive index.

Referee comment: "Figure 8 What kind of a scale is on the abscissa. The minor tick marks cannot be used to state what the precision is exactly, but it appears to be less than 3% for 532 nm. This figure then indicates that a difference in aerosol radius between the min and max distribution width, which is about a factor of two in radius, lead to almost no effect on extinction? This is a surprising result, suggesting that the determination of the median radius is not that critical. There is a much larger effect at 1064 nm but it seems a bit odd that the effect is not symmetric."

Our response: Thank you for pointing out the obviously distorted Figure, we have corrected it. In section 4 we added a short explanation about the relative error used as abscissa scale in Figure 9 (since we added one figure in the revised manuscript this corresponds to Figure 8 of your comment). The influence of the distribution width has of course an influence but it is not that big. The differences between the two wavelengths are caused by a wavelength dependant sensitivity. The scattering cross section and the extinction coefficient are not linear functions of the radius. Therefore, a symmetric behaviour can not be expected here.

Referee comment: "254-258 According to Eq (11) the only size distribution information

used is the median radius, r_m, which appears in the phase function. Thus I don't follow this argument that the uncertainties in distribution width are compensated for by the opposite uncertainty in median radius. The uncertainty in distribution width leads to the radius uncertainty which is then used in calculating extinction, according to Eq (11)."

Our response: If the assumed distribution width is too large, i.e. larger than its real value, then the retrieved median radius will be low biased and vice versa. These effects partly compensate each other for the determination of the aerosol extinction coefficients. For this reason, the extinction coefficient retrievals react less sensitively to an erroneous distribution width compared to the median radius retrievals.

Referee comment: "Figure 9 What is the point of this figure It is just a repeat of Figure 3 with the per cent uncertainties, already shown in Figure 8, added to absolute extinction, and it is much less helpful than Figure 8 in assessing this uncertainty."

Our response: This observation is right. We prefer to separate the retrieved profiles (without error ranges) from the error estimation. In our view this approach improves the understanding.

Referee comment: "Eq (16) c(z) is not defined. Shouldn't there be a ratio of wavelengths within the exponential term of the desired wavelength over the reference wavelength. "

Our response: Thank you for pointing this out, there was no reference on c(z) in the text, we have changed this.

Referee comment: "Figure 11 Which lidar extinction profile is used in the Angstrom conversion to the satellite wavelengths. "

Our response: This information is now included.

Literature used:

Craig F. Bohren and Donald R. Huffman, "Absorption and Scattering of Light by Small Particles", Wiley Science Paperback Series, Wiley Professional Paperback Edition 1998

Jäger, H., T. Deshler, and D. J. Hofmann, Midlatitude lidar backscatter conversions based on balloonborne aerosol measurements, Geophys. Res. Lett., 22, 1729-1732, 1995.

McLinden, C. A., McConnell, J. C., McElroy, C. T., and Grif?oen, E.: Observations of stratospheric aerosol using CPFM polarized limb radiances, J. Atmos. Sci., 56, 233-240, 1999

Mie scattering routines, University of Oxford, Departement of Physics, http://eodg.atm.ox.ac.uk/MIE/index.html, last access: 31 December 2018

Bourassa, A. E., Degenstein, D. A., and Llewellyn, E. J.: Retrieval of stratospheric aerosol size information from OSIRIS limb scattered sunlight spectra, Atmos. Chem. Phys., 8, 6375-6380, 2008

Ugolnikov, O. S. and Maslov, I. A.: Investigations of the Background Stratospheric Aerosol Using Multi- color Wide-Angle Measurements of the Twilight Glow Background, Cosmic Research, 56, 2, 85-93, doi:10.1134/S0010952518020119, 2018, Original Russian Text, 2018, published in Kosmicheskie Issledovaniya, 2018, 56, 2, 95-102, 2018

Please also note the supplement to this comment:
https://www.atmos-meas-tech-discuss.net/amt-2019-267/amt-2019-267-AC3-supplement.pdf